# Developing a quality framework for community pharmacy: a systematic review of international literature

Ali M K Hindi ![ORCID] ,[1] Stephen M Campbell,[2,3] Sally Jacobs,[1] Ellen Ingrid Schafheutle ![ORCID] [1]

[1]Centre for Pharmacy Workforce Studies, Division of Pharmacy and Optometry, The University of Manchester, Manchester, UK
[2]Centre for Primary Care, University of Manchester, Manchester, UK
[3]Department of Public Health Pharmacy and Management, School of Pharmacy, Sefako Makgatho Health Sciences University, Pretoria, Gauteng, South Africa

**Correspondence to**
Dr Ali M K Hindi;
ali.hindi@manchester.ac.uk

## ABSTRACT

**Objective** To identify the defining features of the quality of community pharmacy (CP) services and synthesise these into an evidence-based quality framework.

**Design** Systematic review following the Preferred Reporting Items for Systematic Reviews and Meta-Analyses guidelines.

**Data sources** International research evidence (2005 onwards) identified from six electronic databases (Embase, PubMed, Scopus, CINAHL, Web of Science and PsycINFO) was reviewed systematically from October 2022 to January 2023. Search terms related to 'community pharmacy' and 'quality'.

**Eligibility criteria for selecting studies** Titles and abstracts were screened against inclusion or exclusion criteria, followed by full-text screening by at least two authors. Qualitative, quantitative and mixed-method studies relevant to quality in CP were included.

**Data extraction and synthesis** A narrative synthesis was undertaken. Following narrative synthesis, a patient and public involvement event was held to further refine the quality framework.

**Results** Following the title and abstract screening of 11 493 papers, a total of 81 studies (qualitative and quantitative) were included. Of the 81 included studies, 43 investigated quality dimensions and/or factors influencing CP service quality; 21 studies assessed patient satisfaction with and/or preferences for CP, and 17 studies reported the development and assessment of quality indicators, standards and guidelines for CPs, which can help define quality.

The quality framework emerging from the global literature consisted of six dimensions: person-centred care, access, environment, safety, competence and integration within local healthcare systems. Quality was defined as having timely and physical access to personalised care in a suitable environment that is safe and effective, with staff competent in the dispensing process and pharmacy professionals possessing clinical knowledge and diagnostic skills to assess and advise patients relative to pharmacists' increasingly clinical roles.

**Conclusion** The emerging framework could be used to measure and improve the quality of CP services. Further research and feasibility testing are needed to validate the framework according to the local healthcare context.

## STRENGTHS AND LIMITATIONS OF THIS STUDY

⇒ This review deployed a comprehensive and systematic search of the international literature, which sought to identify defining features of the quality of community pharmacy healthcare services and synthesise these into a quality framework.

⇒ For data extraction, a two-step selection process was conducted: two authors (AMKH and SMC) screened all 11 493 papers independently of each other, and the two other authors (SJ and EIS) reviewed all papers with discrepancies and/or queries.

⇒ To ensure the relevance of the findings to patients, members of the public who use community pharmacy services were consulted on the findings, and their feedback was used to further refine the dimensions and subdimensions of the quality framework.

## BACKGROUND

Faced with growing patient needs, workforce shortages and financial constraints, the necessity for healthcare systems worldwide to focus on delivering 'high-quality care' and meeting demand for primary care has never been greater, with evidence of wide variation in quality between and within countries.[1 2] Health policy in the past few decades has focused on measuring and improving the quality and safety of healthcare services,[3] as well as improving the quality of care via a wider workforce approach (ie, distribution of clinical responsibilities between professions) and local integration of health and social care globally.[4] The aim is to improve and strengthen a quality health and care system by joining up the planning, commissioning and delivery of health and care services to provide seamless, locally based integrated care that meets people's needs promptly and effectively.[3 5 6]

In relation to this, in the past two decades, policymakers have increased the range of

healthcare services provided by community pharmacies (CPs), over and above their more traditional medicine supply function, to relieve burden on general medical practice and expand capacity within primary care systems.[7] CPs are accessible and convenient, offering extended and weekend opening hours. Unlike other primary care providers, patients can access CPs without the need for an appointment. Hence, CPs are well-positioned to improve patient access to care and may assist in reaching patients in deprived areas.[8]

With a view to increasing patient access and choice, healthcare systems worldwide, most notably in countries such as the UK,[9 10] Canada,[11] USA,[12] Australia[13] and New Zealand,[14] have invested in expanding the range of healthcare (ie, medicine-related and public health) services offered by CP alongside the sale of over-the-counter (OTC) medicines and other items. However, the quality of some CP services, for example, dispensing and medication review services, has been inconsistent.[15–17] Given the increasing range and volume of services provided by CP, it is important to consider how the quality of care can be improved and made equitably accessible. To be able to assess the quality of healthcare provided by CP, an agreed-upon definition and framework are needed.[16]

Different definitions and frameworks of healthcare quality have emerged across healthcare over the years. One of the most influential models stems from Donabedian's structure–process–outcome framework (1980).[18] 'Structure' involves the setting of care (eg, physical facility, human resources and equipment), 'process' encompasses the actions taken during service provision (eg, diagnosis and treatment), and 'outcome' is the result of actions taken (eg, clinical changes to health and patient satisfaction). Donabedian proposed that structure, process and outcomes are closely linked and influence each other, and his three components are the basis for many quality frameworks.[19–22]

In 2001, the US Institute of Medicine (IOM) developed a healthcare quality framework that involved six dimensions (ie, safety, effectiveness, patient-centredness, timely, efficient and equitable).[23] The IOM's framework has been widely recognised, and since its inception, different organisations have proposed quality frameworks that often use a combination of these six dimensions. Notably, the Organisation for Economic Cooperation and Development (OECD) Health Care Quality Indicators Project (2006)[24] and Lord Darzi's Next Stage Review (2008)[25] defined quality under the three dimensions of safety, effectiveness and patient-centredness. More recently, similar to the IOM's quality framework but also acknowledging the importance of integration, the WHO Framework on Integrated People-centred Health Services (2018) described high-quality care as care that is safe, effective, people-centred, timely, efficient, equitable and integrated.[3]

Since the early 2000s, definitions of quality in healthcare have been developed and continue to be refined. However, quality is still not well defined in CP,[15 26] and little is known about what quality in CP means or how to measure it.[26] In 2012, Halsall *et al* characterised healthcare quality in UK CP under three dimensions: 'accessibility, 'effectiveness' and 'positive perceptions of the experience'.[27] More recently, Watson *et al* characterised quality under the dimensions of person-centredness, professionalism and privacy.[28–30] A US-based study looking at patients' understanding of what constitutes a 'quality pharmacy' identified themes focusing on patient care and trust in pharmacists.[31] However, the dimensions of quality proposed in these studies were mainly related to pharmacists' more traditional role in medicine supply. Furthermore, these studies did not seek to develop a quality framework for CP health service provision as part of an integrated primary healthcare system. As the expansion of CPs away from a primary medicine supply role and into an extended range of professional services gathers pace,[32] there is a need to shed light on ways CPs could work effectively with other primary care providers to provide better-quality healthcare services.

CP provides an exemplar of a (partly) publicly funded private sector provider in a mixed-market healthcare system. Similar to CP, quality is poorly defined in other private sector primary care providers such as dentistry[22 33] and optometry.[34] As stated in the WHO report, *'For if quality of care is not ensured, what is the point of expanding access to care?'*.[1] In line with the policy drive to increase patient choice and access to a wider range of services and service providers, it is important to develop a better understanding of quality in these sectors.[10 35]

> "We cannot assess quality until we have decided with what meanings to invest the concept. A clear definition of quality is the foundation upon which everything is built (Donabedian, 1985)".

The aim of this study is to identify the defining features of the quality of CP services and synthesise these into an evidence-based CP quality framework.

## METHODS

This systematic review is reported according to the Preferred Reporting Items for Systematic Reviews and Meta-Analyses statement.[36]

### Search strategy

Six electronic databases were searched (ie, Embase, PubMed, Scopus, CINAHL, Web of Science and PsycINFO) using search terms relating to 'community pharmacy' and 'quality' (table 1). Specific search strategies for each database are provided in online supplemental file 1. Database searches were reviewed with the University of Manchester library's team. In addition, references to the included studies were scanned for further relevant studies. The search strategy included studies published between 2005 and January 2023. The date limitation, set from 2005 onward, corresponds to the initiation of the revised pharmacy contract in the UK, which is at the forefront of

**Table 1** Search strategy

| Concept | Key terms* |
|---|---|
| Healthcare quality | 'Quality' OR 'healthcare quality' OR 'quality of healthcare' OR 'quality improvement' OR 'quality assessment' OR 'quality assurance' |
| AND Community pharmacy | 'Community pharmacy' OR 'retail pharmacy' |

*Different wildcards and truncations were used depending on the database.

international developments, introducing novel pharmacy services and advancing pharmacist roles.

### Data screening

A two-step selection process was conducted by two reviewers (AH and SC) independently of each other (conventional double screening). Non-English papers were translated. Titles and abstracts were initially screened against the inclusion and exclusion criteria by AH and SC, followed by subsequent full-text screening (table 2). During the double-screening process, two additional reviewers (SJ and ES) were consulted where there was a discrepancy between AH and SC and/or queries arose.

### Data extraction and synthesis of results

Data from the included papers were extracted using NVivo as a data extraction grid. The process of synthesising the literature was iterative. The first author (AH) initially catalogued the different dimensions and theoretical concepts of quality arising from the literature. Data

relevant to the quality of CP healthcare services generated from the literature were then categorised across these identified dimensions of quality. All authors independently assessed each dimension. Iterative revisions were made based on discussions between all authors.

A narrative synthesis was then undertaken by the first author to provide a descriptive account of both qualitative and quantitative research evidence. Synthesis involved integrating and drawing on findings from studies that addressed quality dimensions, factors influencing the quality of CP healthcare services, and factors influencing the integration of services with the wider healthcare system. Synthesis also involved studies that developed quality indicators and standards for CP and studies that assessed patient satisfaction with and/or preferences for CP when they provided findings relevant to the aim of the review. As the focus of this review was to synthesise findings into dimensions that are relevant to quality, findings emerging from the data from different methodological approaches were combined to contribute to an emerging quality framework.

### Critical appraisal

As the included articles used qualitative, quantitative or mixed-methods approaches, different methodological quality assessment tools were employed. Qualitative studies were assessed using the JBI checklist for qualitative research. The tool consists of a 10-point checklist, each requiring a response of 'yes' (1), 'no' (0), 'unclear' (0) and 'not applicable'.[37] Cross-sectional studies were assessed using the JBI checklist for cross-sectional studies. The tool consists of an eight-point checklist, each requiring a response of 'yes' (1), 'no' (0), 'unclear' (0) and 'not applicable'.[38]

**Table 2** Inclusion and exclusion criteria

| Inclusion criteria | Exclusion criteria |
|---|---|
| **Setting:** Community pharmacy | Non-community pharmacy setting |
| **Design/study type:** Empirical studies | **Design/study type:** Literature reviews |
| **Location:** All regions | |
| **Publication date:** 2005 onwards | |
| **Publication type**: Peer-reviewed journal papers Reports on QI indicator development | **Publication type:** Conference abstracts Commentary, opinion pieces and editorials Reviews |
| **Focus of study:** ► Definitions and dimensions of quality in community pharmacy (including patient experience, environment and safety) ► Development and assessment of quality indicators and standards for community pharmacy healthcare services ► Patient satisfaction with community pharmacy healthcare services ► Factors influencing quality of care in community pharmacy | **Focus of study:** ► Advancing the scope of pharmacists and/or pharmacy technicians in practice ► Integrating pharmacists or pharmacy technicians in other healthcare settings ► Pilot community pharmacy interventions and services ► Evaluations of individual services ► Impact of training ► Evaluations of pay-for-performance schemes ► Assessing approaches to measure quality (eg, quality inspection reports, quality cards and administrative claims) |

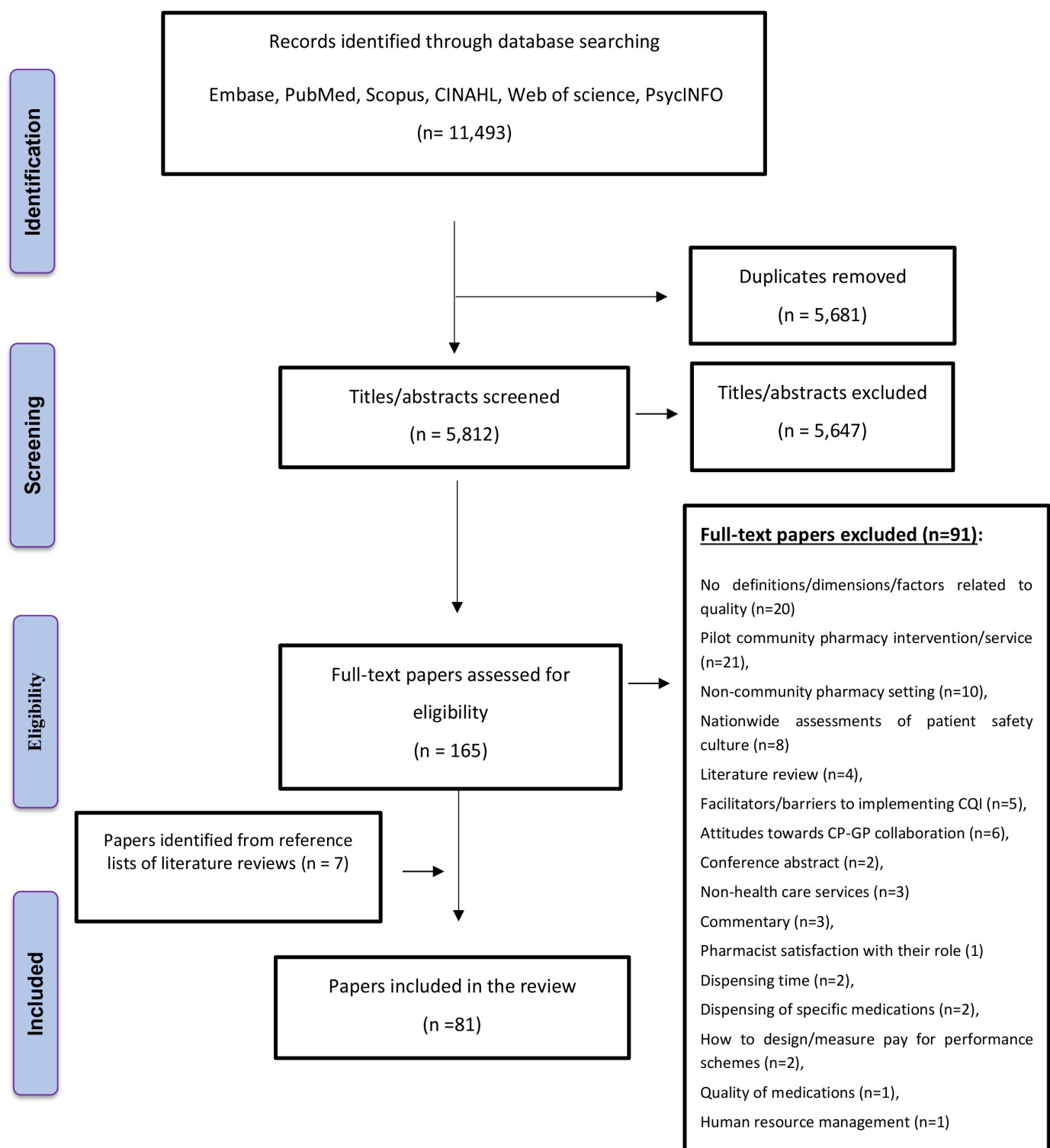

**Figure 1** Flow diagram demonstrating the search procedure.

The Mixed-Method Appraisal Tool was employed to evaluate mixed-method studies, enabling the assessment of their methodological quality. Seventeen criteria were considered, each requiring a response of yes (1), no and cannot tell (0).[39] The Conducting and Reporting Delphi Studies checklist was utilised for Delphi studies. It is important to highlight that this checklist primarily serves as a reporting tool rather than a methodological one. Nonetheless, for consistency, we employed a criterion to assess the nine items on the checklist (yes=1, no and cannot tell=0).[40]

Quality assessment checklist selection was done by AH and SC. The quality assessment process was carried out by AH, who has conducted quality appraisals for two previously published systematic reviews. The overall quality of the literature was evaluated based on the total score

for each checklist. Studies were not excluded based on quality, but the score helped to critically appraise findings. Total scores are reported without classification of the studies based on specific quality thresholds, as the authors of these tools did not suggest cut-offs.

## Patient and public involvement

Following the synthesis of findings, an online patient and public involvement event was held in April 2023 with seven members of the public who use CPs. These members were recruited via patient charity organisations, where the lead author provided a summary of the study with contact details for dissemination. This event was held to ensure the incorporation of the patient perspective in ongoing discussions about quality initiatives in CP. At the event, a summary of findings was presented by the lead author under the dimensions of the quality framework. Following the presentation, members of the public were asked:

► Do the initial findings make sense?
► Does the 'person-centred care' dimension cover the important aspects of quality in CP?
► Is there anything important missing from the framework in general?

The event gathered feedback on the dimensions and subdimensions of the quality framework emerging from the review. The lead author took notes on the discussion, summarised key points and sent them to participants via email to ensure all important information was captured. Any additional suggestions provided by participants via email were considered. The feedback provided was used to further refine the dimensions and subdimensions of the quality framework.

## RESULTS

### Study selection

A total of 11 493 papers were identified for initial screening after duplicates had been removed. Following title and abstract screening, 165 papers were assessed for eligibility via full-text reading, with 74 studies included in the review. Manual searching of reference lists identified seven additional studies after eligibility screening (figure 1).

### Definition of pharmacy services

Multiple terms were used in the literature to describe aspects of CP practice and healthcare service provision. For consistency, we have broken down CP healthcare services into (1) medicines supply and (2) professional pharmacy services (table 3).

Some medicines are available to buy without a prescription, commonly referred to as OTC medicines. Data from studies which focused on sale and supply (be that on prescription or in response to a request for sale) of OTC medicines were grouped under 'medicine supply'. Data from studies which looked at the sale and supply of OTC medicines involving professional or clinical judgement, for example, as part of a service, were included under 'professional pharmacy service'.

### Study characteristics

Of the 81 studies included in the review, 43 investigated quality dimensions and/or factors influencing the quality of CP services.[15 26–30 41–77] Twenty-one studies assessed patient satisfaction with and/or preferences for CP.[78–98] Thirteen studies reported the development or assessment of quality indicators for CPs.[99–111] Four studies described and defined standards or guidelines for good pharmacy practice which can be used to help define quality.[112–115]

Multiple methods were used including: surveys (n=46),[26 41 44 46 47 50 56 58 60 62 64 66–68 74 76 78–82 84–95 98–100 102 105–111 114 115] qualitative interviews (n=9),[15 30 45 49 61 69 70 73 83] focus groups,[27 53 63 77] premeasurement and postmeasurement of adherence to standards,[113] biographic and photographic techniques,[42] participant observations,[51] nominal group technique,[43] applying indicators in practice,[103 104] Q-methodology,[96 97] stakeholder event,[112] deductive content analysis,[71] patient stories[65] and mixed methods (n=4).[48 54 59 72] The remaining studies used two or more qualitative methods (n=5)[28 29 52 57 75] and two or more quantitative methods (n=2).[55 101]

| | |
|---|---|
| **Table 3** Definition of pharmacy healthcare services | |
| Medicine supply | *'The time between when the prescription is received by the pharmacy and the prescribed medicine(s) is supplied to the patient'.*[127] The dispensing process involves: ► Receiving and validating the prescription ► Assessing and reviewing the prescribed medicine ► Selecting/preparing, packaging and checking the medicine ► Labelling ► Supplying and counselling the patients ► Recording the intervention.[128] |
| Professional pharmacy services | *'A professional pharmacy service is an action or set of actions undertaken in or organised by a pharmacy, delivered by a pharmacist or other health practitioner, who applies their specialised health knowledge personally or via an intermediary, with a patient/client, population or other health professional, to optimise the process of care, with the aim to improve health outcomes and the value of healthcare'.*[129] |

Most of the studies were from the UK (n=15),[15] [26–30] [42] [54] [56] [65] [77] [90] [100] [106] [111] USA (n=11)[41] [47] [51] [53] [64] [66] [68] [72] [81] [88] [98] and Australia (n=7).[45] [46] [49] [52] [59] [83] [113] Of the remaining studies, four each were from Japan,[57] [61] [62] [79] the Netherlands[55] [102–104] and Thailand[67] [69] [95] [105]; three each from Germany,[58] [75] [108] Estonia,[71] [78] [115] Iran[48] [76] [84] and Vietnam[93] [96] [97] and two each were from Lebanon,[112] [114] UAE,[60] [86] Brazil[107] [110] and Spain.[63] [73] One each from Canada,[74] Finland,[99] New Zealand,[43] Lithuania,[44] Malaysia,[87] Poland,[91] Slovenia,[80] Serbia,[70] Sudan,[94] Nigeria,[109] Iraq,[92] Pakistan[89] and China.[85] One study involved five European countries (Denmark, Germany, Netherlands, Poland and Great Britain) to validate a pan-European questionnaire.[50] One study was conducted among three African countries: Ethiopia, Uganda and Zimbabwe,[101] and another compared questionnaire findings between CP users in Poland and the UK.[82]

Most of the literature explored the views and expectations of CP staff[15] [26] [27] [29] [42] [44–50] [54–56] [60] [61] [64] [66] [68] [69] [77–80] [99] [102] [103] [106–111] [115] and patients.[26–28] [41] [47] [48] [52] [53] [55] [56] [58] [59] [62] [63] [65] [67] [69] [70] [77–87] [89–98] [106] General practitioners' (GPs) views on quality in CP were explored in seven studies.[43] [48] [54] [57] [77] [80] [106] The views of pharmacy organisations and primary healthcare funders and policymakers were explored in just seven studies.[15] [27] [29] [30] [43] [56] [108] [114] Five studies which developed quality indicators explored the views of pharmacy academics.[99] [100] [102] [108] [110] Summary of study characteristics is provided in online supplemental file 2, where they are ordered chronologically.

## Critical appraisal of studies

Nine studies were excluded from critical appraisal as their methods were outside the remit of the quality assessment checklists. These included Q methodology,[96] [116] survey tool user guide,[47] assessment of indicator validity through a systematic framework[71] [101] [103] [104] [113] and a scientific committee meeting for guideline development.[112]

Of the 72 studies that were critically appraised, cross-sectional quantitative studies scored an average of 61%, qualitative studies scored an average of 75%, Delphi studies scored an average of 72% and mixed-method studies scored an average of 76% (online supplemental file 3). However, most cross-sectional studies did not investigate confounding factors. Furthermore, only three[30] [73] [77] of the 21 qualitative studies reported on the influence of the researcher on the research (ie, reflexivity). While the methods used for all studies were appropriate, only three[100] [102] [108] of the nine Delphi studies fully described the stages of the Delphi process, including a preparatory phase, the actual 'Delphi rounds', interim steps of data processing and analysis and concluding steps. Furthermore, two of the four mixed-method studies excelled in only one aspect of the mixed-method design. For example, Snyder et al[72] achieved high quality in the qualitative elements but demonstrated limitations in the quantitative domain. In contrast, Dadfar et al[48] scored high in the quantitative aspect but lacked in the qualitative dimension.

## Quality framework

Data relevant to identifying concepts and dimensions of quality of care for CP identified from the literature were synthesised and themed under six dimensions (access, environment, competence, person-centred care, safety and integration) to develop a quality framework (figure 2). The narrative synthesis below is themed under these six dimensions.

## ACCESS: STRUCTURAL AND PROCEDURAL COMPONENTS OF QUALITY SUCH AS OPENING HOURS, WAITING TIME, PHYSICAL ACCESS, AVAILABILITY OF MEDICINES AND AVAILABILITY OF PHARMACY STAFF TO PROVIDE SERVICES

### Opening hours

Availability of pharmacy services during stated and extended opening hours are commonly identified as key features of quality in CP.[48] [58] [62] [67] [70] [79] [89] [90] [96–98] [110] Patients, pharmacists and GPs suggest that CPs should aim to offer extended opening hours outside regular hours.[27] [28] [30] [53] [77]

### Waiting time

Minimal waiting time for pharmacy services (particularly for picking up medicines dispensed on prescriptions) is commonly cited as an important procedural feature of quality of care in CP.[27] [48] [65] [96] [97] Studies exploring the views of patients on quality of care in CP suggest that pharmacies should aim to minimise wait times to get medicines dispensed.[65] [70]

### Physical access

Five studies describe 'parking space near the pharmacy' as a feature of quality in CP.[28] [70] [78] [80] [89] Three studies highlight the importance of CPs being accessible for people with special needs such as the elderly, visually impaired and people with baby carriages.[70] [80] [110] Ease of access of CPs via public transportation,[58] [85] work/home[53] [96] and other healthcare facilities are important features of quality as perceived by patients.[70] [79]

### Availability of pharmacy staff

Having adequate numbers and appropriately qualified pharmacy staff is described as a hallmark characteristic of a quality CP.[53] [59] [70] [89] [92] [98] [105] Studies commonly measure the availability of a pharmacist (on-site) to provide advice and answer medication-related queries.[89] [92] [105] [115] The availability of pharmacy staff on the phone is addressed in two studies.[62] [70]

### Availability of medicines

Studies in this review indicate that pharmacies should hold an adequate, well-managed stock of medicines as well as medical devices.[61] [79] [112] Studies also emphasise on pharmacies having a stock management system that helps control stock orders and expiry dates and using contingency plans for purchases in an emergency.[79] [103] [105] [109] [114] Furthermore, CPs should have available records for expired drugs, as well as having specific procedures for disposal of expired products.[95] [101] [103] [109] [114]

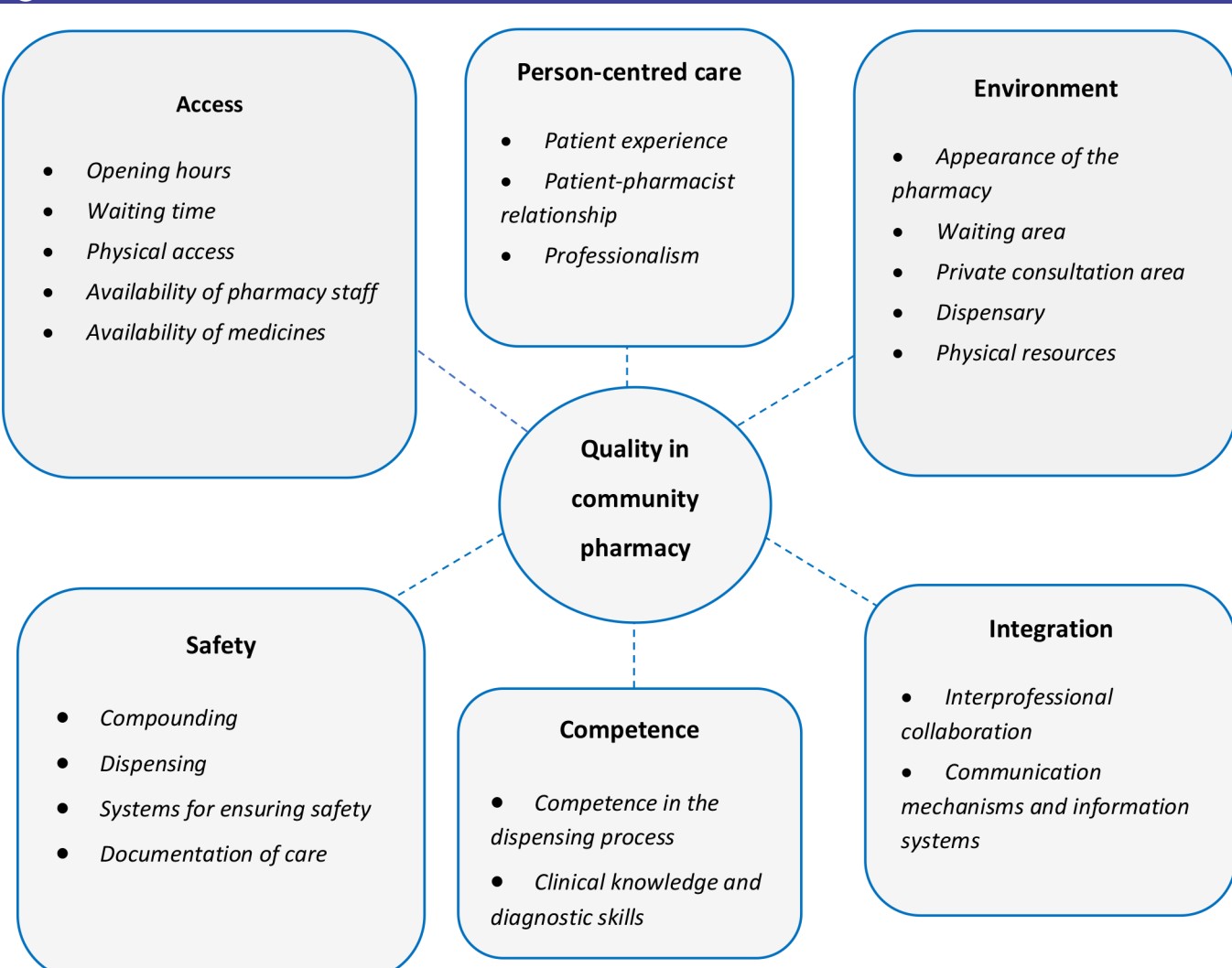

**Figure 2** Overview of quality dimensions emerging from the literature.

Patients, pharmacists and GPs highlight the importance of pharmacies maintaining adequate stock and/or being able to obtain medicines quickly, to avoid patients having to return.[26 30 59] Patients also perceive reasonable and affordable cost of medications and notification of discounts as an important determinant of CP service quality.[53 58 70 83 85 88 89 91 93 97] Patients expect pharmacists to provide them with information about alternative medicines and their prices.[96 97]

### ENVIRONMENT: THE IMPACT OF FACILITIES, EQUIPMENT AND PHARMACY LAYOUT ON THE QUALITY OF HEALTHCARE SERVICE PROVISION

#### Appearance of the pharmacy

The appearance of the CP is an important structural feature of quality health service provision. Studies suggest that CPs need to appear health service orientated by clearly displaying medicines and informational material (such as adverts and leaflets).[80 81 95] The pharmacy should also be positioned in a manner which is visible and accessible to patients with clearly defined boundaries. In supermarkets, it should be clear where the general shop or supermarket ends and the pharmacy begins.[42]

Studies also highlight that every pharmacy should have sufficient counters for dispensing medicines[97] and adequate physical space for pharmacy staff to provide professional services (health promotion, education, consultation or screening services to individuals or groups).[42 105] It is also important to ensure that premises are tidy[48] and lighting of the pharmacy is well distributed.[60]

Cleanliness and hygiene of the pharmacy are commonly highlighted as a feature of quality of care.[59 85 89 91 93 97 105 112] A few studies specifically mention 'ensuring room or air temperature is appropriate'[60 69] and 'avoidance of unpleasant smells'[28 59 69 80] as a means to promote a good first impression of the pharmacy.

#### Waiting area

Studies suggest that a good quality pharmacy should ensure that the waiting area has sufficient space and seating.[54 63 78–80 84 97 115] The importance of informing

patients of waiting times and the reasons for any delays was addressed in one study.[54]

### Dispensary

Studies suggest that the dispensary should be well organised and spaciously designed to ensure efficient processing of prescriptions.[42] Storage shelves or drawers should be clearly labelled with drug classifications and medicines should be kept according to the drug classifications.[105] Pharmacies are required to have a system in place to prevent unauthorised access into areas where controlled drugs are stored.[105 113]

### Physical resources (equipment)

Studies highlight the importance of having drug information systems and resources to ensure the provision of high-quality services.[27 48 57 58 67 112 113] Only two studies specifically mention resources needed to provide professional pharmacy services, such as scales, digital blood pressure monitoring equipment, finger tips and sugar equipment.[95 105]

### Private consultation area

Having a private area for consultations is perceived to be a key facilitator for overcoming privacy issues.[28 30 42 43 52 53 60 63 77 89 90 93 95–97 113–115] Pharmacies without a designated consultation room increase the risk of patient conversations being overheard.[30 52 60 90 96 97] Pharmacies in countries such as the UK are required to have at least one dedicated consultation room, and it is noted that pharmacists should be proactive in offering it to patients.[30 42 77] Relative to pharmacy size, where possible, the room should be spacious, ensuring it is clutter-free and gives the impression of a professional consultation room.[42 77]

### COMPETENCE: OF PHARMACY STAFF IN THE DISPENSING PROCESS, PHARMACY PROFESSIONALS' CLINICAL KNOWLEDGE AND DIAGNOSTIC SKILLS TO ASSESS AND REFER PATIENTS
#### Competence in the dispensing process

Pharmacists' ability, knowledge and expertise (ie, competence) to deliver counselling on prescription medicines are often used to describe the quality of health service delivery in CP.[27 29 30 43 48 57 59 61 67 69 81 92 93 95 97 98 104 109 110 112 113] Patients and community pharmacists suggest that providing high-quality care requires pharmacists having knowledge and skills to dispense the most effective medicines and provide accurate, clear and complete information for a specific medicine.[15 28 30 61 69] Studies also commonly mention speed of dispensing,[89 109] accuracies of dispensing,[70 84 86 89 101 103–105 114] and gathering essential patient information as elements of an effective dispensing process.[55 89 95 103–105 114]

### Clinical knowledge and diagnostic skills

Only four studies (three of which looked at OTC consultations and one at home care supply) describe competence as knowledge and skills which extend beyond traditional dispensing and medicine supply and are particularly relevant for pharmacists' increasingly clinical roles and professional pharmacy services. These studies emphasise the need for pharmacists to have knowledge of specific disease areas[61] and diagnostic skills to provide effective treatment options with correct instructions for medicine usage and storage.[30 57 69] Moreover, GPs expect pharmacists providing professional services to be competent to assess and refer patients to a GP or other healthcare provider if necessary.[30 43]

Some studies highlight pharmacy staff needing more opportunities to enhance clinical knowledge via participation in training programmes, CPD courses and/or seminars.[15 46 79 105 109 112 115] Making use of all the skill sets of employees (ie, skill mix) was suggested as important for improving the quantity and quality of professional services in CP.[15 43 77] Upskilling pharmacy technicians to free up pharmacists to move from medicine supply to professional pharmacy services was suggested in one study.[64]

### PERSON-CENTRED CARE: PHARMACY STAFF PROVIDING PATIENTS WITH A POSITIVE PATIENT EXPERIENCE, ESTABLISHING A PATIENT–PHARMACIST RELATIONSHIP AND DEMONSTRATING PROFESSIONALISM AT ALL TIMES
#### Patient experience

Many studies identified in this review highlight the importance of a positive patient experience when looking at the quality of care in CP. A positive patient experience is often described by patients as pharmacists taking the time to understand patients' individual needs and involving patients in decisions around their medications.[15 30 41 43 44 53 62 67 70 77 93 96 97 106 110] This includes tailoring the delivery of services to people with special needs or minority groups,[53 70 110] for example, by *'adjusting the tone of voice when addressing patients with hearing difficulty'* or *'using capital letters on written materials if) the patient has vision problems'*.[70] Patients, pharmacists and GPs perceived sole trader (independent) CPs to provide more personalised care compared with pharmacy chains due to greater pharmacist autonomy in the former.[15 30 64 91]

### Professionalism

The professionalism shown by pharmacy staff was perceived by patients as a hallmark feature of good quality service provision. Professionalism encompasses attributes such as courtesy, empathy and trustworthiness.[28 48 53 57 58 61 65 67 70 80 81 83 90–93 97 109 113] Studies suggest that patients expect pharmacy staff to treat them with courtesy and respect and spend as much time as necessary during each encounter.[53 57 65 70 80 81 83 90–92 97 113] However, patients perceive a lack of empathy shown by pharmacy staff to reduce service quality.[53 70] Patients valued pharmacists expressing honest opinions regarding patient benefit as a high priority.[58 67 109] In terms of professional appearance, two studies suggest that pharmacists should

be distinguishable from the rest of the staff, for example, by wearing a name badge with their role.[28 105]

### Patient–pharmacist relationship

Studies investigating the views of patients, pharmacists and GPs on CP frequently cite the patient–pharmacist relationship as an important feature of service quality. Trust, friendliness or helpfulness and the availability of the pharmacist have been found to influence the quality of the patient–pharmacist relationship as perceived by patients.[29 41 53 57 59 63 67 77 83] Continuity of care (ie, patients seeing the same pharmacist over time) is perceived to facilitate the development of trust and rapport between patients and pharmacists.[26 29 30 83]

### SAFETY: IDENTIFYING ERRORS AND INTERVENING, ACCURACY IN DISPENSING AND COMPOUNDING, ADEQUATE INFORMATION SHARING BETWEEN PHARMACY STAFF WHEN EXCHANGING SHIFTS AND HAVING SYSTEMS FOR ENSURING SAFETY
#### Compounding

Studies suggest labelling of compounded preparations (ie, preparation of a custom medication) with detailed instructions and clear expiry dates,[70 112] as well as the availability of standard operating procedures (SOPs) to ensure accuracy in compounding.[102–104]

#### Dispensing

Studies commonly mention ensuring the accuracy of dispensing so errors are prevented.[70 84 86 89 101 103–105 114] Identifying and resolving dispensing errors is also seen as a key characteristic of good-quality health service provision in CP. This requires pharmacies to have clear SOPs for checking prescriptions and dispensing medications (particularly high-risk medications).[103 104 106] Studies also suggest having protocols and guidelines for asking patients about potential drug contraindications and drug–drug interactions.[102–104 109]

#### Systems for ensuring safety

Recording prescription data and patient information on computer systems to avoid errors and safety incidents is mentioned in the included papers.[101 104] The literature also suggests that pharmacies should have an internal quality and safety management system in place for registering errors made during dispensing, evaluating patient experiences and recording the number of patient complaints.[102–104 109] Three studies also highlight the importance of investigating and learning from incidents, education and training about safety, staffing and management commitment to patient safety.[43 47 50]

#### Documentation of care

Studies looking at the documentation of patient care focus on the accurate recording of relevant information, such as medical history and medication,[30 61 67 112–114] in a way that can be read and interpreted by other healthcare professionals.[110] Furthermore, these studies measure

whether patients' personal information is stored and disposed of in confidential manner.[60 61 88]

One study measured handovers defined as 'exchange of information, responsibility and accountability when a pharmacist concludes a shift and another replaces them at the beginning of a new shift within the same pharmacy'.[66] The study identified that almost half of the time, handoffs that occur in a CP setting are inaccurate or incomplete.[66]

### INTEGRATION: WAYS FOR CP TO ESTABLISH AND SUSTAIN RELATIONSHIPS WITH THE WIDER HEALTHCARE TEAM BY HAVING INTERPROFESSIONAL COLLABORATION, COMMUNICATION MECHANISMS AND INFORMATION SYSTEMS
#### Interprofessional collaboration

The ability of community pharmacists to establish a relationship with the local GP was perceived as a fundamental part of CP integration with the wider healthcare system.[15 54 72 73 77] Building a relationship required a shared understanding of competencies, roles and responsibilities.[74 75 77] The perceived benefit of having closer CP–GP working relationships was improved communication, effective signposting and prompt resolution of prescription issues,[15] handling near-misses and dispensing errors, and ensuring errors and near misses are recorded and disused regularly.[103 104 106]

#### Communication mechanisms and information systems

GPs' and community pharmacists' preference for communication methods (eg, telephone and face-to-face) has been explored but findings are inconclusive.[74 76] One study highlights that pharmacists express a preference for predefined and clear ways to communicate with GPs, given difficulties getting GPs on the phone and receiving an answer to their query.[75] Having a lead responsible for linking GP and CPs is suggested in one study as a potential way to facilitate CP–GP collaboration.[73]

Whether the CP should have not only read but also written access to shared medical records has been debated. This would allow pharmacists to view relevant information about a patient's medical history to inform their assessment and clinical judgement and enable them to add prescription and medical or intervention details in the patient's medical record, so doctors and the wider general practice team are aware.[30 57 73 74 76 77 88 90] Pharmacists, in some studies, argue they require better access to patient information to provide safe and effective healthcare services.[74 76 77 88] Equally, in the UK, patients and GPs have raised concerns over read-and-write access to medical records, considering the sharing of patient information with commercial organisations, with limited control over who has access, as problematic.[30 74 77]

Three Commonwealth studies highlight the importance of having shared communication systems between CP and the rest of the healthcare system to facilitate CP integration.[43 65 77] In one of these studies, GPs argue that

it is difficult to refer patients to CP given that interactions at CP are not documented or communicated to them.[77]

## FRAMEWORK REFINEMENT BASED ON PATIENT AND PUBLIC INVOLVEMENT

When members of the public were presented with findings and asked for input on dimensions and subdimensions of the quality framework emerging from the review, most were dissatisfied with waiting times at CP to collect their medicines. There were tensions around the only pharmacist on site not being accessible to patients.

In addition, the CP retail environment was perceived as a barrier to good quality service provision, mainly due to privacy issues (eg, asking for details such as address and date of birth in front of customers). All members highlighted the importance of CP staff being professional and distinguishable by wearing a name badge with their role.

Furthermore, integration was seen as a key element of quality, and members described the lack of collaboration or communication between GPs and pharmacists. Lastly, members of the public mentioned that CPs are unaware when patients are directed towards them by GPs and vice-versa. This input from the patient and public involvement group was used to further refine the dimensions and subdimensions of the quality framework (online supplemental file 4).

### Definition of quality of care in CP

Based on the findings in this review, quality of care in CP can be defined as having timely and physical access to person-centred professional services in a suitable environment that is safe, integrated and effective. Key dimensions in this review linked to Donabedian'structure–process–outcome components are summarised in online supplemental file 5.

## DISCUSSION

In the absence of a universally agreed quality framework looking at health service provision in CP, this review aimed to collate and synthesise concepts explored in the literature that are relevant to defining quality of care in CP. On synthesising the findings of 81 papers, quality was conceptualised by the interrelated dimensions of person-centred care, access, environment, competence, safety and integration.

The dimensions of quality identified in this review resonate with the IOM's six dimensions of quality,[23] the OECD's proposed definition of quality[24] and the WHO framework on integrated people-centred health services.[117] The dimensions common to all frameworks were person-centeredness, effectiveness, access and safety. In line with the WHO framework, the framework developed here for quality in CP also included an integration dimension, whose importance and relevance for CP are discussed below.[3 118] Unlike these other frameworks, however, 'environment' was conceptualised as a separate dimension. Lack of privacy in CP was commonly highlighted by this review as a barrier to providing high-quality healthcare services. The 'shop' appearance of CPs and whether premises are fit for purpose may prohibit some CPs from meeting all aspects of the framework.[42] One way of being able to ensure privacy when appropriate (eg, for professional services) is to have a dedicated consultation area with adequate space.[119]

This review, which adopts a broad view of features of quality of care in CP, draws out important considerations for defining quality CP to ensure high-quality patient care, experience and outcomes. To begin with, CP is one of the most accessible settings in which to receive healthcare services.[17] However, geography alone does not guarantee that patients will receive the healthcare services they need. Corroborating findings from this review, previous literature reviews suggest that improving access further involves having adequate staffing levels, strategies for managing medicine supply as well as shortages and efficient workflow procedures to reduce waiting times.[8 120–122]

The responsiveness of health systems to the needs of the population is a central pillar of healthcare quality and a crucial perspective is through patients' evaluations of the care they receive.[123] In line with findings from the wider primary and secondary care literature,[124 125] the person-centred care dimension in this review highlights a positive patient experience, a good patient–pharmacist relationship, relational continuity of care and professionalism as key attributes of quality from a patient perspective. A systematic review looking at a wide range of primary and secondary care settings found that patient experience is positively associated with clinical effectiveness and safety.[124] Moving forward, quality initiatives in CP need to prioritise collecting patient feedback, with an emphasis on organisations using that data as one aspect of ongoing quality improvement.

In this review, the competence dimension mainly covered pharmacy staff's ability to effectively perform the dispensing procedure, with dispensing remaining a significant part of CPs, even where (funded) professional services are emerging. Although many studies did not cover professional services, much of the medicine supply process is now expected to be performed by the pharmacy support team, which is an important part of freeing pharmacists' time for professional services. As the scope of professional CP services continues to expand in many countries, more research is needed to develop quality indicators that consider pharmacy professionals' clinical knowledge and diagnostic skills for providing professional (clinical and public health) services.

The dimensions of access, person-centred care, competence and environment mirrored those of existing CP frameworks by Halsall[27] and Watson.[28–30] However, compared with previous studies conceptualising quality in CP, the 'integration' dimension was unique in our framework. Six studies synthesised in this review and patient and public involvement members describe CP integration within the wider healthcare system as an

important dimension of a quality framework. Our study suggests that an integration dimension needs to consider interprofessional collaborations and information sharing between CP and other primary care providers, such as general practice. The 'interprofessional collaboration' element of our integration dimension resembles Valentijin's taxonomy of integrated primary care,[126] where the term 'professional integration' is used to describe '*interprofessional partnerships based on shared competencies, roles, responsibilities and accountability to deliver a comprehensive continuum of care to a defined population*'. The communication mechanisms and information systems of our integration dimension closely align with Valentijn's 'functional integration', defined as '*key support functions and activities (ie, financial, management and information systems) structured around the primary process of service delivery to coordinate and support accountability and decision-making between organisations and professionals to add overall value to the system*'.[126]

To the authors' knowledge, this is the first systematic review of the international literature that sought to identify defining features of the quality of CP healthcare services and synthesise these into a quality framework. The framework emerging from this review contributes to knowledge of improving access to, and the healthcare of, the population through privately owned businesses that provide publicly funded primary healthcare services. The strength of this paper is the comprehensive and systematic search of the international literature deployed by the lead author (AH) with conventional double screening by an expert in quality of care (SC). Furthermore, an expert in CP policy research (ES) reviewed all papers at the full-text review stage, where there were disagreements and uncertainty between AH and SC. Another expert in CP policy research (SJ) undertook this process on all papers where discrepancies remained. Moreover, input from public contributors was used to further refine the dimensions and subdimensions of the quality framework. In terms of limitations, only one author critically appraised the findings due to time constraints. Given that this review sought to develop a broad framework covering different dimensions of healthcare quality, the word 'integration' was not used as a keyword in the search strategy, which could explain the low number of papers identified relative to integration.

## CONCLUSION

This review defines the quality of CP and provides a dimensional framework for the quality of CP services consisting of six dimensions: patient experience, access, environment, safety, competence and integration. As CP expands in the UK and other countries beyond a primary medicine supply function, the quality dimensions need to be validated and refined locally, with a particular emphasis on integration. Integration is particularly relevant for professional services, where roles and responsibilities for joined-up services are shared across primary care providers, making collaboration and two-directional information sharing particularly important. Once quality dimensions are validated and refined, the next step will be using the framework to develop and feasibility test summative 'quality assurance' and formative 'quality improvement' mechanisms.

**Acknowledgements** We would like to thank the National Institute for Health and Care Research (NIHR) School of Primary Care for funding the fellowship. We would also like to thank our patient and public contributors for providing their input on the quality framework emerging from this systematic review.

**Contributors** AH, EIS, SMC and SJ conceptualised the study. AH ran database searches, title and abstract screening. AH and SMC undertook independent full-text review. EIS reviewed all papers at the full-text review stage, where there were disagreements and uncertainty between AH and SMC; SJ undertook this process on all papers where discrepancies remained. All authors discussed and agreed inclusion and exclusion criteria, and judgements on all papers at the full-text review stage, to reach a final decision. AH ran the data extraction process which was refined by SMC, SJ and EIS. AH facilitated the patient and public involvement event, which SJ co-facilitated. AH wrote the first full draft of the manuscript. All co-authors reviewed and discussed drafts iteratively, providing critical and intellectual contributions to analysis, interpretation and framing. Guarantor, AH.

**Funding** This work was funded by The National Institute for Health and Care Research (NIHR) School of Primary Care (Grant reference:C066)

**Competing interests** None declared.

**Patient and public involvement** Patients and/or the public were involved in the design, or conduct, or reporting, or dissemination plans of this research. Refer to the Methods section for further details.

**Patient consent for publication** Not applicable.

**Ethics approval** Not applicable.

**Provenance and peer review** Not commissioned; externally peer reviewed.

**Data availability statement** Data are available upon reasonable request.

**ORCID iDs**
Ali M K Hindi http://orcid.org/0000-0002-1076-435X
Ellen Ingrid Schafheutle http://orcid.org/0000-0001-7072-0888

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
