## [Reviewer comments · BMJ Open]

ARTICLE DETAILS

TITLE (PROVISIONAL)	Developing a quality framework for community pharmacy: A systematic review of international literature
AUTHORS	Hindi, Ali; Campbell, Stephen; Jacobs, Sally; Schafheutle, Ellen

VERSION 1 – REVIEW

REVIEWER	Hamde Nazar Newcastle University
REVIEW RETURNED	27-Sep-2023

GENERAL COMMENTS	Many thanks for the opportunity to review this work. It has been conducted and reported to a very high standard. Please can the authors justify the date limitation on the search set at 2005? The use of the PPI event is really interesting. More detail on how and why this event was undertaken and how 'data' was captured would be really helpful. There is wider discussion about how researchers use and report on PPI, so the approach used would be of interest. Otherwise, I have no reservations in recommending this manuscript for publication.
--

REVIEWER	Margaret Watson University of Strathclyde, Strathclyde Institute of Pharmacy and Biomedical Sciences
REVIEW RETURNED	21-Oct-2023

GENERAL COMMENTS	This is a large systematic review conducted using standard systematic review methods. It is both timely and comprehensive and the manuscript is well-written. The systematic review is reported using the PRISMA checklist but it is not referenced in the manuscript. A statement should be added to the Methods to confirm that the review complies with and is reported using the PRISMA checklist. The included studies were not critically appraised. Critical appraisal of the included studies should be undertaken and reported and the results reviewed to determine whether their interpretation is altered as a result of the critical appraisal process. Whilst a several methods were reported across the included studies, many used similar methods e.g. cross-sectional design, and as such critical appraisal should be possible and derive meaningful outputs. More information should be included regarding the patient and public participants - who were they, how recruited, and so on. What were
---

	the actual methods used to present the review results to these individuals and what methods (formal or informal) were employed to explore and document their opinions? The results should be presented to reflect the order presented in the sentence on page 10 line 27-29 or the sentence could be re-ordered (for ease). How are the studies ordered in the Table that describes their method/results? It appears that it might be chronological - oldest to most recent. The authors should consider whether the tabulated results could be re-ordered/grouped to provide additional learning i.e. grouped by country or design, or focus etc? A statement should be made in the accompanying text re how the studies are ordered or grouped in the table. Supplementary File 4 - add key to define "type" ie. S, P. Consider removal of terms such as "on the other hand".
--	--

REVIEWER	Beverley Glass James Cook University, Pharmacy
REVIEW RETURNED	07-Nov-2023

GENERAL COMMENTS	This is an excellent paper - a pleasure to read. Thank you A minor correction - page 7 line 27 Data ----- were I am surprised that PPI does not require Ethics as I am not sure this would be the case in Australia. But accept your explanation. My main concern for this review is the lack of quality appraisal for the papers - as per page 4 (Quality of the papers was not critically appraised) and then the inclusion in the limitations of "due to considerably varying methodologies and findings". This is although in the inclusion criteria, the publication type specified. In the systematic reviews in BMJ all I have considered present a quality assessment of papers - using for example the ROBINS-I for Quantitative papers and the JB Critical Appraisal for Qualitative Research. So either including a quality audit or changing the review to a scoping review should be considered. This is very important work and i really look forward to see how this framework progresses and is validated.
---

VERSION 1 – AUTHOR RESPONSE

Reviewer #3	
Comments:	Response
This is an excellent paper - a pleasure to read. Thank you	Many thanks. Really glad you enjoyed reading the manuscript.
A minor correction - page 7 line 27 Data ----- were	We have now implemented this minor correction.
My main concern for this review is the lack of quality appraisal for the papers - as per page 4 (Quality of the papers was not critically appraised) and then the inclusion in the limitations of "due to considerably varying methodologies and findings". This is although in the inclusion criteria, the	We strongly agree with this suggestion and have now conducted a critical appraisal. To accommodate for the varying methodologies, we used multiple checklists for different study types. We did not provide cut-offs as authors of the checklist advised

publication type specified. In the systematic reviews in BMJ all I have considered present a quality assessment of papers - using for example the ROBINS-I for Quantitative papers and the JB Critical Appraisal for Qualitative Research. So either including a quality audit or changing the review to a scoping review should be considered.

This is very important work and i really look forward to see how this framework progresses and is validated.

against this. Furthermore, many of the studies were descriptive in nature. Nevertheless, we included an overall quality statement in the results section along with supplementary file 3, ensuring that all studies and their respective scores are accessible:

(methods: page 7):

“As the included articles used qualitative, quantitative, or mixed methods approaches, different methodological quality assessment tools were employed. Qualitative studies were assessed using JBI checklist for qualitative research. The tool consists of 10-point checklist, each requiring a response of ‘Yes’ (1), ‘No’ (0), ‘Unclear’ (0), ‘Not Applicable’.³⁷

Cross-sectional studies were assessed using JBI checklist for cross sectional studies. The tool consists of an 8-point checklist, each requiring a response ‘Yes’ (1), ‘No’ (0), ‘Unclear’ (0), ‘Not Applicable’.³⁸

The Mixed Methods Appraisal Tool (MMAT) was employed to evaluate mixed methods studies, enabling the assessment of their methodological quality. Seventeen criteria were considered, each requiring a response Yes = 1, No and Cannot tell = 0.³⁹ The Conducting and Reporting Delphi Studies (CREDES) checklist was utilised for Delphi studies. It's important to highlight that this checklist primarily serves as a reporting tool rather than a methodological one. Nonetheless, for consistency, we employed a criterion to assess the 9 items on the checklist (Yes = 1, No and Cannot tell = 0).⁴⁰

Quality assessment checklists selection was done by AH and SC. The quality assessment process was carried out by AH who has conducted quality appraisal for two previous published systematic reviews. Overall quality of the literature was evaluated based on the total score for each checklist. Studies were not excluded based on quality, but the score helped to critically appraise findings. Total scores are reported without classification of the studies based

on specific quality thresholds as the authors of these tools did not suggest cut-offs”.

(Results: page 10)

“Nine studies were excluded from critical appraisal as their methods were outside the remit of the quality assessment checklists. These included Q methodology,^{99 119} survey tool user guide⁵⁰ assessment of indicator validity through a systematic framework,^{74 104 106 107 116} and a scientific committee meeting for guideline development.¹¹⁵

Of the 72 studies that were critically appraised, cross-sectional quantitative studies scored an average of 61%, qualitative studies scored an average of 75%, Delphi studies scored an average of 72%, and mixed methods studies scored an average of 76% (Supplementary File 3). However, most cross-sectional studies did not investigate confounding factors.

Furthermore, only ^{three}_{30 76 80} of the twenty-one qualitative studies reported on the influence of the researcher on the research (i.e. reflexivity). Whilst the methods used for all studies were appropriate, only three^{103 105 120} of out the nine Delphi studies fully described the stages of the Delphi process, including a preparatory phase, the actual ‘Delphi rounds’, interim steps of data processing and analysis, and concluding steps. Furthermore, two of the four mixed methods studies excelled in only one aspect of the mixed methods design. For example, Snyder et al.⁷⁵ achieved high quality in the qualitative elements but demonstrated limitations in the quantitative domain. In contrast, Dadfar et al.⁵¹ scored high in the quantitative aspect but lacked in the qualitative dimension.”